# Metabolomics: An Emerging Approach to Understand Pathogenesis and to Assess Diagnosis and Response to Treatment in Spondyloarthritis

**DOI:** 10.3390/cells11030549

**Published:** 2022-02-04

**Authors:** Chiara Rizzo, Federica Camarda, Denise Donzella, Lidia La Barbera, Giuliana Guggino

**Affiliations:** Department of Health Promotion, Mother and Child Care, Internal Medicine and Medical Specialties, Rheumatology Section, University of Palermo, Piazza delle Cliniche 2, 90110 Palermo, Italy; chiararizzo87@gmail.com (C.R.); federicacamarda@gmail.com (F.C.); denise.donzella@gmail.com (D.D.); lidialb90@gmail.com (L.L.B.)

**Keywords:** spondyloarthritis, metabolomics, psoriatic arthritis, ankylosing spondylitis, microbiota, biomarkers

## Abstract

Spondyloarthritis (SpA) is a group of rheumatic diseases whose pathogenesis relies on a complex interplay between genetic and environmental factors. Over the last several years, the importance of the alteration of the gut microbiota, known as dysbiosis, and the interaction of bacterial products with host immunity have been highlighted as intriguing key players in SpA development. The recent advent of the so called “-omics” sciences, that include metabolomics, opened the way to a new approach to SpA through a deeper characterisation of the pathogenetic mechanisms behind the disease. In addition, metabolomics can reveal potential new biomarkers to diagnose and monitor SpA patients. The aim of this review is to highlight the most recent advances concerning the application of metabolomics to SpA, in particular focusing attention on Ankylosing Spondylitis and Psoriatic Arthritis.

## 1. Introduction

Spondyloarthritis (SpA) refers to a group of chronic inflammatory diseases sharing heterogeneous clinical features and pathogenic mechanisms. It encompasses several distinct diseases; among them, Ankylosing Spondylitis (AS) and Psoriatic Arthritis (PsA) are the most common ones [1].

Recently, the Assessment of SpondyloArthritis International Society (ASAS) has subdivided SpA into axial SpA (axSpA) and peripheral SpA. AS can be considered the prototype of axSpA, while PsA is one of the forms of mostly peripheral SpA. AxSpA primarily affects the spine and the sacroiliac joints; on the other hand, peripheral SpA manifestations are usually arthritis, enthesitis and dactylitis [2].

Moreover, the absence or presence of radiographic sacroiliitis allows for distinguishing the non-radiographic SpA (nr-axSpA) and the radiographic axSpA (r-axSpa), respectively.

The systemic nature of SpA is highlighted by its extra-musculoskeletal manifestations, such as acute anterior uveitis, psoriasis or inflammatory bowel disease (IBD) [3].

According to the ASAS criteria, SpA can be diagnosed in patients with lower back pain for ≥3 months, with onset age <45 years old, sacroiliitis on imaging and at least one other SpA feature, or positivity for HLA-B27 and at least two other SpA features. SpA features consist of inflammatory back pain, arthritis, enthesitis, dactylitis, uveitis, psoriasis, IBD, good response to NSAIDs, family history of spondyloarthropathy, HLA-B27 positive status and elevated C-reactive protein. Patients without current low back pain, but with peripheral arthritis or enthesitis or dactylitis, plus at least one or two of above mentioned SpA features, are classified as affected by “peripheral SpA” [2].

Global estimation of axSpA prevalence ranges from 0.5% to 1.5% showing geographic differences that can be explained mainly by the prevalence of the HLA-B27 [4].

Nr-axSpA is equally common in men and women, whereas AS is more common in men, with a male to female ratio of 2 to 3:1, and it is strongly associated with HLA-B27 positivity. Data about real prevalence and incidence of peripheral SpA are still lacking, but a recent meta-analysis reported pooled prevalence rates of 22.9% for arthritis, 13.6% for enthesitis and 5.6 % for dactylitis in AS and nr-axial SpA patients [5].

The pathogenesis of SpA relies on a complex interplay between genetic factors, with HLA-B27 representing the strongest associated gene, and several environmental triggers.

Among environmental factors, a growing body of evidence supports the strict correlation between SpA and subclinical gut inflammation, which has been reported in approximately 60% of patients with SpA [6]. In these patients, the progression to IBD has been observed in up to 10% [7].

Chronic inflammatory gut changes principally occur in AS and PsA and are characterised by mononuclear cell infiltrate in the lamina propria, villous atrophy and goblet cells hyperplasia, paralleling IBD histological features [8,9].

Increasing evidence supports the occurrence of gut dysbiosis in SpA, although it is still unclear whether it may occur before or after subclinical gut inflammation.

The gut microbiota has been extensively characterised in SpA, and a reduction in biodiversity has been described in both AS and PsA, especially when IBD-like histological features are present. In particular, in AS, an abundance of five families of bacteria (*Lachnospiraceae, Ruminococcaceae, Rikenellaceae, Porphyromonadaceae* and *Bacteroidaceae*) and a decrease of *Veillonellaceae* and *Prevotellaceae* [10] were demonstrated. Moreover, Breban et al. identified a significant increase of *Ruminococcus gnavus* in active SpA patients, making this bacteria a potential marker of disease activity [11].

Gut dysbiosis has been linked to the modulation of gut permeability driven by alterations of the gut-epithelial barrier and the gut-vascular barrier. Specifically, our group previously described the presence of adherent and invasive bacteria in the ileum of AS patients and their association with gut inflammation and zonulin upregulation. Zonulin expression has been associated with downregulation of tight junctions proteins, such as claudin 1, claudin 4, zonula occludens, occludin and VE-cadherin, resulting in the impairment of gut-epithelial and endothelial barrier functions with consequent increased serum levels of bacterial products [12]. Microbiome and bacterial products are related to inflammation in SpA. Their interactions with immune cells, such as type 3 Innate Lymphoid Cell (ILC3), contribute to the production of IL-23, a pivotal cytokine in SpA pathogenesis [13,14].

Nowadays, the identification of a specific biomarker or a panel of molecules to profile SpA patients, especially in the early stage of disease, is still an unmet need in rheumatology. The delay between disease onset and diagnosis is up to 10 years. In this regard, the development of sequencing technologies and multi “-omics” methodologies may allow a better characterisation of SpA to help early diagnosis and grant a tailored therapeutic approach along with a proper monitoring.

Over the last decade, thanks to the metabolomics, a deeper description of the metabolic profile, at both the serum and tissue level, of AS and PsA has emerged. The metabolic fingerprinting in such diseases could explore even the link between dysbiosis, bacterial metabolites and disease development, representing an intriguing new field of research.

## 2. Metabolomics with a Focus on Rheumatic Disease

The “-omics” sciences comprise a huge family of new technologies that can be applied to medicine and include genomics, transcriptomics, proteomics and metabolomics. The peculiarity of -omics methodologies resides in the non-targeted approach to identify biomarkers that can be useful to stratify patients. A preliminary knowledge of the molecules identified is not required. Such technologies are characterised by a very high discriminatory ability and even small quantities of each tissue are sufficient to conduct a complete analysis [15].

The potential of metabolomic procedure consists in the unbiased identification of small molecules, known as metabolites, in a biologic system. It can easily catch changes of the metabolic status of several tissues and biological fluids that are supposed to reflect disease related perturbations. The analysis of acquired data is conducted through the alignment with an existing database of known metabolites to drive their identification in the samples studied. Metabolomics relies on two main high-throughput technologies: nuclear magnetic resonance (NMR) and mass spectrometry (MS). Usually, after the acquisition of experimental data, a multivariate statistical analysis is performed and, finally, when the metabolic profile is detected, the metabolome, allows for building up the patient map.

It is worth mentioning that both NMR and MS can be applied to every specimen, virtually every biological tissue or fluid. A major advantage of those techniques is that only a minimum amount of samples is required; specifically, 1 mL for liquids and 1 mg for solids can be sufficient [16]. This is of great importance, especially when dealing with difficult-to-obtain samples, such as synovia, enthesal or ileal tissues or synovial fluid, in the absence of abundant joint effusion.

NMR and MS are not exactly interchangeable tools. Specifically, MS is more sensitive but requires a good preparation of the sample through the separation of the single constituents via liquid chromatography (LC) or gas chromatography (GC). This implies a greater variability, that the event depends on the experience of the laboratory, making results less reproducible. On the other hand, NMR does not need a pre-treatment of the specimens, so they can be reused for further analysis, making NMR more reproducible despite a lower sensitivity.

In both cases, the final result is a plot that identifies many metabolites, typically of low molecular weight, accounting for the metabolome of a specific sample [17,18].

As a fast-developing research field, metabolomics has been applied to several medical specialties, such as cardiology, oncology, diabetology and gastroenterology. The detection of new possible biomarkers through this kind of analysis has raised interest even in the field of rheumatology, and this technique has been used to study rheumatic diseases.

Research has been conducted at both preclinical and clinical levels, and the profiling obtained was proven effective in the diagnostic process, in discriminating patients according to disease activity and even in the assessment of response to treatment [19].

Literature data on metabolomics applied to rheumatoid arthritis (RA), osteoarthritis and systemic lupus erythematosus are available. Only a few studies have focused attention on SpA. The purpose of the following paragraphs is to review the current knowledge on the metabolomic profiling in AS and PsA, shedding light on the detection of new possible biomarkers and on the pathogenesis of such diseases.

## 3. Metabolomics Applied to Spondyloarthritis

The application of metabolomics to SpA is still in its infancy as the required techniques have only recently become easily available. The types of samples analysed include plasma, ligaments, urine and faeces. However, a limitation of all studies published dealing with the topic is the small sample size assessed and the lack of independent validation [20].

In addition to this, SpA is still a complex disease to diagnose. Patients experience important delays in diagnosis and, as a result, treatments are often started only after several years.

Up to now, no reliable biomarkers have been identified in SpA, and early diagnosis is still challenging. In this regard, metabolomics offers a new approach, not only to better understand the pathogenetic mechanism behind SpA, but even to allow an early diagnosis and a correct stratification of patients that may drive the choice of a more tailored therapeutic strategy [21] (Figure 1).

Potential pathways explored in SpA highlight a role for gut microbiota-related metabolic profiling, fat and amino-acid metabolism. Moreover, in peripheral arthritis, specific metabolite changes seem able to help in differentiating PsA from RA [22].

### 3.1. Metabolomics in Ankylosing Spondylitis

The quest for a reliable marker of disease in SpA has pushed the application of the multi-“omics” approach even to AS and axSpA. Ideally, the detection of an array of biomarkers, selectively expressed during AS, is the objective of metabolomics.

The first “discovery” study from 2012 by Fischer et al. explored the serum small metabolites in 18 AS patients compared to nine healthy controls. Authors identified a set of 215 compounds differentially regulated between the two groups. In clinical practice, handling such a huge number of variables is challenging and even their identification was not further pursued in the study. However, to refine the analysis, 10 molecules, displaying the highest separating power, were selected and proved to be effective in distinguishing AS from controls. Enlarging the panel of molecular features to 65 elements, it was even possible to categorize AS patients according to disease activity (evaluated by BASDAI). The most interesting result from this study is the identification of a specific metabolic pathway related to AS. In particular, a reduction in the (23*S*,25*R*)-25-hydroxyvitamin D_3_ 26,23-peroxylactone, a metabolite of 25(OH)D_3_ produced in the kidney, in the AS cohort was found [23]. This suggests a role for the alteration of vitamin D3 metabolism in AS that, as known, is characterised by dysregulated immune response and altered bone remodelling. Vitamin D is implied in both functions and an impairment in its metabolism may account for a possible pathogenetic role in AS.

In 2016, the first NMR spectroscopy in AS based on a multiple tissue approach on plasma, urine and hip ligaments was published. Plasma and urine were obtained from 44 patients and 44 controls and hip ligament samples from 30 patients and 30 controls. Twenty metabolites, able to discriminate between the two groups, were identified. The metabolic pathways were related to fat metabolism, glucose metabolism, intestinal microbial metabolism, choline metabolism and immune regulation pathways. Interestingly, the metabolites described were not shared across the different tissues examined, pointing out that a tissue specific metabolic response in AS exists.

Analysing fat metabolism, a decrease in triglycerides, coupled with an increase in glycerol and β-oxidated fatty acids (acetoacetate, acetone, 3-hydroxybutyrate), was noted [24]. The latter compounds derive from triglyceride metabolism and account for an increased fat metabolism switch in AS that seems related to increased energy demand due to chronic inflammation. Such observations were confirmed in another study by Gao et al. [25]. Moreover, to strengthen this result, fat derived metabolites were increased in urine. On the other hand, branched-chain amino-acids deeply related to fat metabolism, namely leucine and valine, were reduced in plasma. Even if, with contrasting results, an alteration in amino acids level was differentially described in AS and RA compared to healthy controls [26], its implication in energy homeostasis is yet to be established [27].

On ligament tissue, the triglycerides content was dramatically increased, demonstrating an aberrant fat deposition at this level. Additionally, choline was found to be decreased in ligaments. Choline normally prevents fat deposition in tissues and promotes the use of fatty acids. Its reduction at the tissue level in AS may contribute to ectopic fat deposition phenomena [28]. At the urine level, four markers related to intestinal microbial metabolism were found to be altered, suggesting an imbalance in the gut microbiota. In particular, glycine and hippurate were elevated, whereas phenylacetylglycine and butyrate decreased. The link between AS development and dysbiosis may then be reflected by variations in the levels of those metabolites. Lastly, plasma levels of betaine, methionine and N-acetyl glycoprotein (NAG) were increased in AS [24]. Both betaine and methionine are implied in the production of methionine enkephalin (MENK), a regulator of immune response. Their increase suggests an impairment in MENK functions and partly justifies the immune system dysregulation observed in AS [29]. In addition, NAG is renowned for its role in promoting glycoprotein acetylation, a fundamental process for white blood cell recognition. An increase in NAG level may account for leucocytes’ alterations in AS [30,31]. In this regard, the reduction of taurine, an amino-acid that participates in membrane stabilisation and controls oxidative stress and immune system hyperactivation, corroborates the importance of aberrant metabolic pathways in AS development [32].

The lipid and protein breakdown, with a consequent increase in fat metabolism and catabolic activity, has been confirmed in a recent paper that used NMR to study serum samples from 81 SpA patients [33]. Not only did authors identify a panel of metabolite able to discriminate patients from controls, but they further tried to relate metabolites fluctuations to disease activity. Zhou et al. had already demonstrated a very strong correlation between lysine, proline, alanine, serine and clinimetric indexes, such as BASDAI, ASDAS and BASFI [26]. More recent observations showed that, especially when using ASDAS, the sera of severely active patients showed an increase in serum levels of acetate, lactate and phenylalanine and a decrease in serum levels of glycerophosphocholine, glutamine, creatine, methionine and creatinine with respect to inactive and moderately active disease [33].

The possibility of adding objective parameters to disease activity assessment may empower rheumatologist performance in detecting disease flares, assessing remission and response to treatment.

Regarding response to treatment, the study by Gupta et al. found the persistence of metabolic switch towards high fat consumption and an increase in NAG level in persistently active patients, despite treatment. Specifically, patients were assessed at baseline and after three months of active therapy. AS who stayed active showed a worsening in the level of NAG, acetate, lactate and glutamate at the second visit. A deeper analysis, considering different drugs used, was not performed, representing a limitation of the study. Moreover, only 17 patients were assessed pre and after treatment, and the cohort is clearly too small to detect more significant variations [33].

A recent study focused attention on metabolic changes related to anti-tumor necrosis factor alpha (anti-TNFα) treatments in AS. Thirty-two patients and 40 controls were enrolled, and LC-MS analysis was performed on sera samples and repeated after 24 weeks of active treatment. A panel of 55 differently expressed metabolites was detected at the time of diagnosis. By the application of a specific statical tool named LASSO (least absolute shrinkage and selection operator), a final set of 5 metabolites able to identify patients versus controls was derived. l-glutamate, arachidonic acid, l-phenylalanine, PC (18:1(9Z)/18:1(9Z)) and 1-palmitoylglycerol are the five molecules comprised in the diagnostic panel. Again, main pathways involved in AS were confirmed to belong to amino-acid biosynthesis, glycolysis, glutaminolysis, fatty acids biosynthesis and choline metabolism. After treatment, the levels of 21 metabolites were found to be restored when comparing responders to healthy controls. The consequent conclusion is that effective treatment may revert an aberrant shift in metabolism in AS [34]. However, the observation deserves to be deeply studied in larger cohorts of patients, and it is worth emphasizing that the authors did not find any difference in the metabolic profile of responders versus non-responders before treatment, meaning that a signature predictive of response to therapy is still far from being discovered.

Interestingly, the hyperactivation of glutaminolysis, as reflected by the increase in glutamate and the decrease in its precursor, glutamine, has been described in cancer and mirrors the energetic abnormal metabolism demonstrated even in AS, which may display some tumor-like metabolic features. Moreover, glutaminolysis is known as a key energy source for Th17 cells, which are major players in AS development and chronic inflammation sustainment [35,36].

The discrimination between mostly peripheral or axial SpA has been attempted without satisfactory results. In fact, only *N*-acetyl glycoproteins levels, commonly raised in inflammation, appeared discriminative and were increased in peripheral SpA without reaching statistical significance [33]. This suggests that maybe the inflammatory burden is increased when peripheral manifestations are present. Further investigation on this topic is required to draw more detailed conclusions.

As described before, dysbiosis in the faecal microbiota has been identified in AS compared to controls [37,38]. Intestinal metabolites are at the interface in the crosstalk between bacteria and intestinal cells. According to this, faecal and gut samples potentially represent good matrices for metabolomic studies, and their alteration may reveal specific AS characterisation.

In this regard, a first essay based on a GS-MS approach identified 19 different small molecules able to cluster AS patients from healthy controls. The AS “faecal signature” involved higher concentrations of 5-trimethylsilyloxy-n-valeric acid, cyclohexanecarboxylic acid, gluconic acid, serine, α-l-galactofuranoside and *N*-(4,5-dimethyl-thiophen-2-yl)-benzamide and lower concentrations of β-sitosterol, 24-ethyl-δ (22)-cloprostenol and 3-pyridinecarboxylic acid [39]. Metabolites were related to imbalanced lipid and glucose metabolism, decreased anti-oxidation activity, steroid hormones unbalance and gut microbiota alteration. However, the extent to which the differences retrieved were related to microbial alteration were not further analysed in the paper, and results were not confirmed in other NMR studies. Using NMR, eight metabolites were identified as biomarkers in the faecal metabolome of AS patients. Two of these, methionine and butyrate, were also described in the study on plasma, hip ligament and urine [40]. For methionine, results appear in the line between the two studies, while opposite findings emerge for butyrate, confirming the difficulties in interpreting the application of metabolomic to a complex, multisystemic disease, such as AS. Investigation on faecal samples in a paediatric population of Juvenile idiopathic arthritis (JIA) and enthesitis-relates arthritis (ERA) demonstrated alterations in the tryptophan metabolism pathway in patients compared to controls [41]. Tryptophan was confirmed as a potential biomarker of AS even in a very recent study that applied metabolomics via LC-MS to intestinal biopsies. In AS tissues, a reduction in tryptophan, paired with an increased in indole-containing metabolites, emerged [42]. Tryptophan has anti-inflammatory properties and a growing body of evidence supports a role in homeostasis of gut epithelial integrity and immune cell functions [43,44]. Tryptophan is exclusively derived from food, and human cells cannot metabolize it into indole derivatives. Gut bacteria express tryptophanase and can generate indole metabolites that have recently been observed in human cancer cell lines [45]. The hyperactivity of this pathway may justify the reduction of tryptophan in AS plasma. In addition to this, the decrease in tryptophan may be justified even by the induction of IDO (indoleamine 2,3-dioxygenase), which converts it to kynurenine, determined from interferon-gamma (IFN-γ) activation in AS [46]. Interestingly, the alteration in the intestinal tryptophan metabolism was not related to a specific bacterial dysbiosis but was rather associated with a dysbiotic community effect, as already proposed for axSpA [42].

To sum up, metabolomics is a promising approach to identify biomarkers in AS, even if only preliminary and not validated results have been reported so far. The possibility of exploring several tissues and fluids is fascinating and will allow a deeper characterisation of AS pathogenesis as well as the identification of a feasible panel of biomarkers for early diagnosis and effective follow up.

### 3.2. Metabolomics in Psoriatic Arthritis

PsA is an extremely heterogeneous disease with several different manifestations occurring within the same patient. Up to now, the early identification and the monitoring of PsA remain a challenge for rheumatologists.

Epidemiological data showed that 80% of PsA patients present PsO before developing articular disease and 30% of patients with pure PsO evolve into PsA [47].

As AS, PsA is not a classical autoimmune disease. There are no specific antibodies and reliable biomarkers have not been identified yet.

In the last decade, few metabolomic studies concerning PsA have been published. Most of the research was indeed focused on psoriasis (PsO) [48]. The two diseases share a common background but present specific features at different levels, such as genetics, transcriptomics and proteomics. Here, we will review the most relevant studies on PsA, including new evidence on the transition from PsO to PsA and the differential metabolomic profile between these conditions.

Regarding diagnosis, metabolic profiling of serum, plasma and peripheral blood mononuclear cells (PBMC) has depicted alterations specific to PsA. In particular, PsA patients present higher glucuronic acid levels compared to healthy controls. When compared to PsO, PsA shows a lower level of alphaketoglutaric acid and an increased level of lignoceric acid, allowing a differentiation between the two diseases [49]. In plasma, tyramine appears to be increased in PsA while mucic acid decreased [50]. Considering lipid metabolism, a disturbance in phospholipid and polyunsatured fatty acid (PUFA) was demonstrated in psoriatic disease. Two studies showed a reduction in fatty acids and, at the same time, an increase in lipid peroxidation products and endocannabinoids in both PsO and PsA. The reduction in fatty acid was significantly higher in PsA patients [50,51].

In PBMC, specifically among lymphocytes, Wójcik et al. found higher 8-isoPGF2a and 4-HNE in PsA, while PsO showed increased 4-HNE adducts, once again demonstrating an alteration of lipid peroxidation processes. Among eicosanoids, which collectively were demonstrated to be increased in PsO and PsA, accounting for a pro-inflammatory environment, two molecules are differentially expressed in the two diseases. Particularly, 15-d-PGJ2 and 15-HETE were elevated in PsA versus PsO [52].

In addition, metabolomics could offer new insights into the differential diagnosis between types of arthritis of recent onset. In the early phases of disease, the only presence of peripheral arthritis may be challenging, and PsA often resembles RA at the beginning. By comparing PsA and RA metabolites, a differential level of lipids, organic compounds and amino-acids has been demonstrated. Madsen et al. retrieved higher aspartic acid, glutamic acid, glutamate, histidine, serine arachidonic acid, cholesterol, threonic acid and 1-monooleoyglycerol and lower glutamine, heptanoic acid, succinate, psudouridine, inosine, guanosine, arabitol, cystine, cysteine and phosphoric acid in PsA [53]. A second study, from 2020, comparing PsA to seronegative RA confirmed such results, also finding a reduced level of phenylalanine in PsA [54]. It is worth underlining that, especially in the absence of autoantibodies, the possibility of clearly distinguishing between PsA and RA represents an important goal to drive the most appropriate treatment. In fact, even if there is an overlap between treatments for these two chronic inflammatory diseases, deep differences exist and some drugs are effective only for one kind of arthritis. This mirrors the activation of different pathways related to inflammation and a precise profiling of patients could help clinicians with choosing between therapies.

The assessment of disease activity is another key point in PsA. The detection of metabolites to measure it was attempted in a few studies. Trimethylamine-N-oxide (TMAO), a well-known risk factor for cardiovascular disease and obesity, positively correlated with PsA activity at both the skin and joint levels, assessed by Body Surface Area (BSA) and Disease Activity Score 28 (DAS28) and Clinical Disease Activity Index (CDAI), respectively [55].

Fluctuations of eicosanoids were proven to be related to PsA activity too. As expected, several pro-inflammatory eicosanoid such as PGE2, 12-oxo-HETE, HXB3 or 6,15-dk-,dh-,PGF1a correlated with disease activity parameters (tender joint count, swollen joint count, CDAI, SDAI and DAS28). Interestingly, some anti-inflammatory eicosanoids, such as 11-HEPE, 12-HEPE and 15-HEPE, showed a positive correlation with the same indexes. In addition, some anti-inflammatory eicosanoids, resolvin D1 and 17-HDoHE, were lower in patients presenting high disease activity. It comes out clearly that an imbalance in lipid profile and eicosanoids level participates in PsA inflammation. More studies are needed to identify a stricter pathogenetic role for these metabolites [56].

Patients with higher disease activity showed a reduced citrate level in urine samples, as demonstrated by Kapoor et al. in 2013 and confirmed in a subsequent study [57,58]. Urine is an easy-to-collect biological sample; its collection is not invasive and analysis can be repeated over time. For these reasons, identifying a reliable urine biomarker for PsA can improve the approach to diagnosis and follow up.

Multiple matrices studies that aim to combine metabolomic profiles from different tissues from the same patient still lack PsA.

However, metabolic biomarkers, mainly including lipids and amino-acids, may improve PsA diagnosis and disease activity assessment [54].

It is worth noting that TMAO derives from microbial metabolism of choline that takes place in the gut, underlining a link between gut microbiota and PsA activity. On the other hand, amino acids can be the result of increased protein catabolism. In PsA, chronic inflammation is an energy demanding process and the increase in lactate and creatine reflects the hyperactivation of aerobic glycolysis and catabolism [59,60]. Moreover, these metabolites can directly affect the immune system: lactate is able to promote CD4^+^ T cell differentiation towards a proinflammatory phenotype while creatine boosts T cell proliferation and cytokine release [61].

To conclude, further research to identify a clear, reproducible, easy panel of molecules in PsA is needed.

## 4. Conclusions and Future Perspectives

Metabolomics is an under-explored approach that is promising for identifying biomarkers in SpA [62]. In this review, we have summarised studies that have examined metabolites profiling in patients with AS or PsA using NMR or MS for several different biological specimens. Such techniques are not easily available and require high laboratory and statistical expertise to be effectively performed.

At present, there is a clear lacking in the standardisation of the technique used (NMR or MS) and the samples collection methods (e.g., fasting or not fasting blood collection), so that studies are not easily comparable. Moreover, in most cases, only small cohorts are analysed.

The majority of studies identified amino-acids and fat metabolism alterations in relation to disease development, diagnosis and activity. The role of the gut microbiota as a potential key player in immune dysregulation in SpA has emerged even through metabolomic studies.

Further research in the field is needed to identify and validate metabolomic biomarkers that can accurately and reliably predict SpA evolution, differentiate patients from controls and assess disease activity.

## Figures and Tables

**Figure 1 cells-11-00549-f001:**
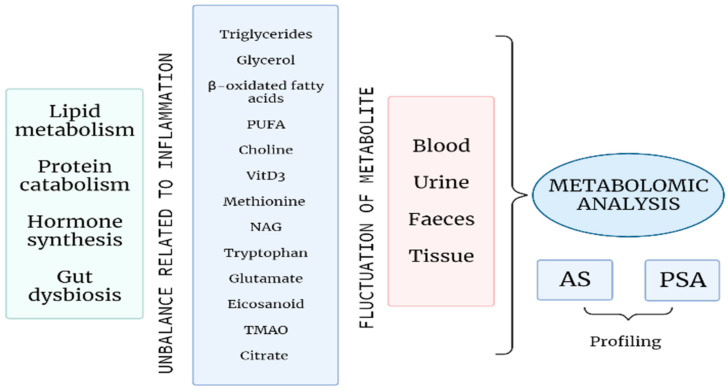
A profound alteration of the energetic metabolism, with an increase in lipid metabolism and protein catabolism; an unbalanced hormone synthesis, mainly affecting bone remodelling processes; and gut microbial dysbiosis are related to inflammation in the course of SpA. Several metabolites, mirroring these processes, undergo fluctuations during SpA and can be identified in different biological samples. The metabolomic analysis of such tissues and fluids can determine a specific profiling of patients with AS or PsA that may allow early diagnosis and personalised treatment decision and monitoring. (PUFA: Polyunsatured fatty acids; VitD_3_: Vitamin D_3_; NAG: *N*-acetyl glycoprotein; TMAO: Trimethylamine-N-oxide; AS: Ankylosing spondylitis; PsA: Psoriatic arthritis).

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
