# Peer review of "Metabolomics: An Emerging Approach to Understand Pathogenesis and to Assess Diagnosis and Response to Treatment in Spondyloarthritis"

_cells, 2022, doi:10.3390/cells11030549_

Round 1

Reviewer 1 Report

Very Interesting Review in an important topic. Congratulations

Author Response

Reviewer 1 comment: Very Interesting Review in an important topic. Congratulations

Response:

We thank the Reviewer for the kind comment. This review deals with a new topic in Spondyloarthritis and we think that it will be an interesting future research field in rheumatology.

Reviewer 2 Report

The review presented by the authors "Metabolomics: an emerging approach to understand pathogenesis and to assess diagnosis and response to treatment in Spondyloarthritis" addresses a topic of special importance and relevance in the field of Spondyloarthritis. It is a very complete review, properly presented and with a number of references consistent with the objective of the review.

Author Response

Reviewer 2 comment: The review presented by the authors "Metabolomics: an emerging approach to understand pathogenesis and to assess diagnosis and response to treatment in Spondyloarthritis" addresses a topic of special importance and relevance in the field of Spondyloarthritis. It is a very complete review, properly presented and with a number of references consistent with the objective of the review.

Response:

We thank the Reviewer for the comment. To the best of our knowledge, we summarized all the most interesting papers dealing with metabolomics and Spondyloarthritis, trying to even underline the role of gut dysbiosis in this perspective. We think that metabolomics will be an important field of research  in rheumatology.

Reviewer 3 Report

This paper is rather a paper indicating the perspectives of metabolomics in studying chronic inflammatory conditions such as spondyloarthritis, and not a true review. However, as a future perspective paper it should be much shorter. 

Author Response

Reviewer 3 comment: This paper is rather a paper indicating the perspectives of metabolomics in studying chronic inflammatory conditions such as spondyloarthritis, and not a true review. However, as a future perspective paper it should be much shorter. 

Response:

We thank the Reviewer for the comment. We reviewed the most interesting and recent literature dealing with metabolomics applied to Spondyloarthritis, specifically addressing this topic in Ankylosing Spondylitis and Psoriatic Arthritis. Up to date no many papers have been published on this issue and we tried to detail what is known about this topic. We think that among “-omics” sciences, metabolomics stands out as a promising research field to deeply understand SpA pathogenesis, to identify new biomarkers of disease and to approach patients from a different point of view.

Reviewer 4 Report

The manuscript is well written and summarized. I have the following comments.

1) Line 12. In accordance with Line 26, are --> is 

2) Line 20-21. (AS) and (PsA) need not be added, for it is not reintroduced in the abstract.

3) Line 32. PsA cannot be stated as the main form of peripheral SpA. It has several manifestations by itself. The sentence should be revised, such as 'one of the forms of...'

4) At the end of figure 1 legend, the acronyms used in the figure should be spelled out (PUFA, VitD3, NAG, TMAO, AS, PsA).

Author Response

Reviewer 4 comment: The manuscript is well written and summarized. I have the following comments.

1) Line 12. In accordance with Line 26, are --> is 

2) Line 20-21. (AS) and (PsA) need not be added, for it is not reintroduced in the abstract.

3) Line 32. PsA cannot be stated as the main form of peripheral SpA. It has several manifestations by itself. The sentence should be revised, such as 'one of the forms of...'

4) At the end of figure 1 legend, the acronyms used in the figure should be spelled out (PUFA, VitD3, NAG, TMAO, AS, PsA).

Response:

We thank the Reviewer for the comment. We revised the manuscript as suggested.

1) Line 12. “are” was rectified with “is”

2) Line 20-21. (AS) and (PsA) were deleted

3) Line 32. The sentence was revised as suggested by the Reviewer

4) In the legend of figure 1 the spelling of the acronyms used (PUFA, VitD3, NAG, TMAO, AS, PsA) was added as suggested